# Chronic Low Grade Inflammation in Pathogenesis of PCOS

**DOI:** 10.3390/ijms22073789

**Published:** 2021-04-06

**Authors:** Ewa Rudnicka, Katarzyna Suchta, Monika Grymowicz, Anna Calik-Ksepka, Katarzyna Smolarczyk, Anna M. Duszewska, Roman Smolarczyk, Blazej Meczekalski

**Affiliations:** 1Department of Gynaecological Endocrinology, Medical University of Warsaw, 00-315 Warsaw, Poland; suchta.katarzyna@gmail.com (K.S.); monika.grymowicz@wp.pl (M.G.); calikowna@wp.pl (A.C.-K.); rsmolarczyk@poczta.onet.pl (R.S.); 2Department of Dermatology and Venereology, Medical University of Warsaw, 00-315 Warsaw, Poland; ksmolarczyk@gmail.com; 3Department of Morphological Sciences, Faculty of Veterinary Medicine, Warsaw University of Life Sciences, 02-787 Warsaw, Poland; anna_duszewska@sggw.edu.pl; 4Department of Gynaecological Endocrinology, Poznan University of Medical Sciences, 60-535 Poznan, Poland; blazejmeczekalski@ump.edu.pl

**Keywords:** polycystic ovary syndrome, insulin resistance, chronic inflammation, interleukins, CRP

## Abstract

Polycystic ovary syndrome (PCOS) is a one of the most common endocrine disorders, with a prevalence rate of 5–10% in reproductive aged women. It’s characterized by (1) chronic anovulation, (2) biochemical and/or clinical hyperandrogenism, and (3) polycystic ovarian morphology. PCOS has significant clinical implications and can lead to health problems related to the accumulation of adipose tissue, such as obesity, insulin resistance, metabolic syndrome, and type 2 diabetes. There is also evidence that PCOS patients are at higher risk of cardiovascular diseases, atherosclerosis, and high blood pressure. Several studies have reported the association between polycystic ovary syndrome (PCOS) and low-grade chronic inflammation. According to known data, inflammatory markers or their gene markers are higher in PCOS patients. Correlations have been found between increased levels of C-reactive protein (CRP), interleukin 18 (IL-18), tumor necrosis factor (TNF-α), interleukin 6 (IL-6), white blood cell count (WBC), monocyte chemoattractant protein-1 (MCP-1) and macrophage inflammatory protein-1α (MIP-1α) in the PCOS women compared with age- and BMI-matched controls. Women with PCOS present also elevated levels of AGEs and increased RAGE (receptor for advanced glycation end products) expression. This chronic inflammatory state is aggravating by obesity and hyperinsulinemia. There are studies describing mutual impact of hyperinsulinemia and obesity, hyperandrogenism, and inflammatory state. Endothelial cell dysfunction may be also triggered by inflammatory cytokines. Many factors involved in oxidative stress, inflammation, and thrombosis were proposed as cardiovascular risk markers showing the endothelial cell damage in PCOS. Those markers include asymmetric dimethylarginine (ADMA), C-reactive protein (CRP), homocysteine, plasminogen activator inhibitor-I (PAI-I), PAI-I activity, vascular endothelial growth factor (VEGF) etc. It was also proposed that the uterine hyperinflammatory state in polycystic ovary syndrome may be responsible for significant pregnancy complications ranging from miscarriage to placental insufficiency. In this review, we discuss the most importance evidence concerning the role of the process of chronic inflammation in pathogenesis of PCOS.

Polycystic ovary syndrome (PCOS) is one of the most common endocrine disorders with a prevalence rate of 5–10% in reproductive aged women. It is characterized by (1) chronic anovulation, (2) biochemical and/or clinical hyperandrogenism, and (3) polycystic ovarian morphology. Diagnosis of the syndrome is based according to the Rotterdam criteria (2003), when 2 out of 3 characters is found, while other etiologies are excluded [1]. PCOS has significant clinical implications and can lead to health problems related to the accumulation of adipose tissue, such as obesity, insulin resistance, metabolic syndrome, and type 2 diabetes [2]. There is also evidence that PCOS patients are at a higher risk of cardiovascular diseases, atherosclerosis, and high blood pressure [1,3]. 

Obesity, particularly the visceral type, is very common among PCOS patients, with the prevalence ranging from 38–88% [1]. The results of a meta-analysis conducted by Lim and coauthors indicate that the mean prevalence of obesity among women with PCOS is about 49% [4]. It has been proposed that androgens stimulate the differentiation of pre-adipocytes to adipocytes, especially in the abdomen area, facilitating the development of visceral-type obesity. Obesity is a metabolic condition characterized by chronic inflammation state with higher pro-inflammatory cytokines, chemokines, and oxidative stress (OS) markers levels [5,6]. This inflammatory process is a potential cause of connected with obesity comorbidities, like endothelial dysfunction, atherosclerosis, and coronary heart disease [6,7]. According to known data, inflammatory markers or their gene markers are higher in PCOS patients. Correlations have been found between increased levels of C-reactive protein (CRP), interleukin 18 (IL-18), tumor necrosis factor (TNF-α), interleukin 6 (IL-6), and ferritin in the PCOS women compared with age- and BMI-matched controls [8,9,10]. Other factors include elevated levels of WBC, plasminogen activator inhibitor PAI1, activity of the angiotensin-renin system (RAS) and free fatty acids (FFA). Obese women with PCOS present lower levels of adiponectin and omentin, the cytokines with anti-inflammatory impact and with decreased cardiovascular risk [11,12].

Some studies show correlation of the iron overload with PCOS. High levels of ferritin and transferrin leads to a reduction in anti-inflammatory cytokines and anti-oxidant molecule levels [13]. OS and chronic low-grade inflammation are known to be pathways involved in PCOS pathogenesis.

In this review, we discuss the most importance evidence concerning the role of the process of chronic inflammation in pathogenesis of PCOS.

## 1. Inflammatory Markers

### 1.1. C-Reactive Protein (CRP)

The majority of studies addressing the status of chronic low-grade inflammation in PCOS have focused on the measurement of CRP. It is an acute phase protein produced by the liver following stimulation by interleukin 6 (IL-6) and tumor necrosis factor α (TNF-α). CRP is also directly produced by adipose tissue. Growing evidence supports the concept that CRP may be a marker of the intravascular inflammatory process and is one of the most important predictors for the development of cardiovascular diseases. The first data demonstrating elevation of CRP in women with PCOS come from Kelly et al., who compared only 17 women with PCOS and 14 healthy controls [14]. They found increased serum CRP in the study group and it remained significant when BMI and age were accounted for. These findings were confirmed by a large number of studies. Elevated CRP levels were found by Tola et al., Souza dos Santos et al., Orio et al., and Rudnicka et al. [10,15,16,17]. In the meta-analysis of 31 clinical trials, conducted by Escobar-Moralle et al., they concluded that CRP in women with PCOS is on average 96 % (95% CI: 71–122%) higher than that in control groups [18]. These data have been confirmed, both by a second meta-analysis by Toulis et al., and by studies demonstrating that treatment with metformin causes a significant reduction in CRP levels [19,20]. Evidence suggests that CRP is positively correlated with insulin resistance, body weight, and fatty mass. Nevertheless, there is still uncertainty whether the inflammation is due to the PCOS itself or to insulin resistance and obesity, and more studied are needed to understand the precise mechanism of elevation of CRP concentrations in PCOS women. 

### 1.2. Pro-Inflammatory Cytokines and Chemokines

Cytokines are soluble molecules that are involved in intercellular communication, produced by a wide variety of cells in the body including adipocytes, and being involved in several biological processes which promote the recruitment of macrophages, the proliferation of vascular smooth muscle cells, and their migration from tunica media to tunica intima. They play an important role in the development of atherosclerosis, coronary heart disease, and diabetes mellitus [18,21]. PCOS is associated with elevation of interleukin-18 (IL-18), monocyte chemoattractant protein-1 (MCP-1), and macrophage inflammatory protein-1α (MIP-1α) [21,22,23,24,25]. Il-18 is a proinflammatory cytokine which belongs to the IL-1 superfamily and is closely associated with insulin resistance, metabolic syndrome, and is an important predictor of long-term cardiovascular mortality. It’s concentration is increased in PCOS patients regardless of presence of insulin resistance and obesity, however, obese women with hyperinsulinemia have even higher concentrations of IL-18 [22,23]. Several studies have also revealed that women with PCOS have elevated MCP-1 concentration, which is one of the most studies chemokines playing a major role in the development of atherosclerosis. This observation was verified in PCOS and control group matched for age and BMI [24,25]. PCOS is also associated with an increased concentration of MIP-1α, also known as chemokine (C-C motif) ligand 3 (CCL3), a cytokine involved in recruitment and activation of leukocyte [26,27].

There are many studies devoted to research on importance of interleukin-6 (IL-6), which is a promotor of CRP production in liver. Nevertheless, the evidence has remained persistently ambiguous. There is scientific research showing both significant as well as non-significant association between IL-6 levels and PCOS [28]. The meta-analysis by Escobar-Morreale et al. showed no difference in the serum levels of IL-6 between women with PCOS and controls [18]. Toulis et al., in a meta-analysis of cardiovascular disease risk markers in PCOS women, achieved similar test results [19]. TNF-α is an another cytokine primarily secreted by the visceral adipocytes and well known mediator of insulin resistance. Two meta-analyses of data available have so far not established a clear association between plasma levels of TNF-α and PCOS [18,19]. In subsequent published works, PCOS is also associated with increased levels of other inflammatory markers like nuclear factor kappa ß (NF-kß) and transforming growth factor-ß1 (TGF-ß1) [29]. Franik G. et al. and other authors found significantly lower concentrations of plasma omentin-1 in polycystic ovary syndrome [12]. Omentin-1 is an adiponectin, released mainly by stromal-vascular cells of adipose tissue, which stimulates Akt phosphorylation and increases insulin-stimulated glucose uptake. Additionally, omentin-1 has an anti-inflammatory impact and is associated with decreased cardiovascular risk [12].

Other studies have focused on the association of genetic polymorphism in interleukin genes and their promoters with the phenotype of the syndrome. IL-1a and IL-1b are important mediators of the inflammatory response and are involved in cell proliferation, differentiation, and apoptosis. The results of gene studies are controversial. Kolbus A. et al. detected that IL-1a, but not IL-b, gene polymorphism is associated with polycystic ovary syndrome [30]. Opposite results from Chinese women with PCOS were found by Wang B et al., who did not confirm the association between interleukin-1a gene (IL-1a) c (-889) T variant and PCOS women [31]. Efforts have also been made to investigate possible associations between polymorphisms in the TNF-α gene promotor and PCOS. Results from analysis of nine polymorphisms (-1196C/T, -1125G/c, -1031T/C, -863C/A, -857C/T, -316G/A, -308G/A, -238G/a, -163G/A) in the promoter of the TNF-α gene have exhibited no significant difference in the frequency of any polymorphism between the patients and the control group [32]. Genetic studies on the 031T/C polymorphism were controversial [32]. However, analysis showed that the -308A polymorphism in the promoter of the gene was associated with high androgen concentrations versus elevated TNF-α levels, and the -308A polymorphism seems to be a much more potent promoter of the gene transcription, leading to variations amongst the patients group and [32,33,34].

### 1.3. White Blood Cell Count (WBC)

Chronic inflammatory processes are associated with elevation of white blood cell count. In the large national cohort study by Brown et al., it was concluded that WBC is a predictor of coronary heart disease mortality, independent of smoking and other traditional risk factors. Even modest elevations of WBC are associated with multiple cardiovascular risk factors like increased BMI, adverse lipid profile, and periodontal disease [35]. Elevation of WBC was found by Orio et al., Papalou et al., Herlihy et al., and other authors [17,36,37]. In the study by Tola et al., CRP was statistically significantly higher, but WBC was distributed homogenously between PCOS and the control group [15]. In a study by Rudnicka et al., white blood cells were also statistically significant higher in PCOS than in healthy subjects and correlated positively with androgens, insulin, and BMI [10]. In the studies by Orio et al. and Papalou et al., it was found that insulin resistance and obesity, not hyperandrogenemia, were the main factors which are responsible for increase of WBC among PCOS women. However, other authors demonstrated that androgens are also predictors of leukocyte count in PCOS [10,38]. It was detected that WBC correlated positively with total testosterone, androstendione, DHEAS, and in multiple regression analysis, testosterone was one of the main predictor factors of leukocyte count [10]. It is possible that hyperandrogenemia individually or in combination with central adiposity and IR might explain leukocytosis. The exact mechanism has not been fully elucidated. Thus far, androgen receptors have been identified in lymphoid and nonlymphoid cells of thymus and bone morrow and in various human leukocytes with a particularly high expression in neutrophils [39]. There are also many research studies that indicate therapeutic activity for androgens against human leukemia cell lines in vitro and in vivo [40]. This could provide a possible explanation, that androgen plays an important part in the development and activation of leukocytes and low grade inflammation.

### 1.4. AGEs (Advanced Glycation End-Products) and Oxidative Stress

Advanced glycation end products (AGEs) are highly reactive proteins or lipids mainly derived from nonenzymatic saccharification of reducing sugar on proteins, lipids, and nucleic acids. Several diseases have been associated with increased expression of AGEs such as: Hyperglycemia, renal insufficiency, diabetes, atherosclerosis. Women with PCOS present elevated levels of AGEs [41,42] and increased RAGE (receptor for advanced glycation end products) expression [43].

As studies suggest, AGEs can have an impact on PCOS-related changes in granulosa and theca cell function. Through this mechanism, they can adversely impact steroidogenesis and follicular development. AGEs are connected with hyperandrogenism in PCOS. It is possibly due to changing the activity of numerous enzymes such as cholesterol side-chain cleavage enzyme cytochrome P450, steroidogenic acute regulatory protein, 17α-hydroxylase, and 3β-hydroxysteroid dehydrogenase [41].

AGEs can increase the production of reactive oxygen species (ROS), thereby initiating intracellular oxidative stress. Oxidative stress is identified as a disproportion between oxidization and antioxidation (between production and scavenging of reactive oxygen/nitrogen species), which latterly leads to multiple negative effects on cellular metabolism [44].

A proper amount of ROS is crucial for regulation of transcription factors, expression of apoptosis genes, and antibacterial and anti-inflammatory effects. During pathological conditions, however, when the ROS level exceeds the buffering capacity of antioxidant enzymes and antioxidants, the balance between oxidation and antioxidation shifts the trend to oxidization, resulting in oxidative stress. Mitogen-activated protein kinase (MAPK), tyrosine kinase, Rho kinase, and transcription factor activation can be stimulated by elevated ROS levels, while protein tyrosine phosphatase (PTP) can be inactivated by ROS [45].

In PCOS, many OS markers have been described as elevated, which gives a that OS can lead to an assumption that OS can participate in PCOS patophysiology [46]. Compared with control women, patients with PCOS presented higher circulating concentrations of markers such as homocysteine, asymmetric dimethylarginine, and increased superoxide dismutase activity and decreased glutathione levels and paraoxonase-1 activity [2]. Summary of the inflammatory markers in polycystic ovary syndrome contains Table 1.

## 2. Obesity, Insulin Resistance and Inflammatory Process

The occurrence of PCOS is associated with an increased prevalence of comorbidities such as metabolic disturbances including obesity and insulin resistance. The results of a meta-analysis conducted by Lim and coauthors indicate that the mean prevalence of obesity among women with PCOS is about 49% [4]. What is more, it is known that usually severity of PCOS is aggravated when it is associated with obesity. It is also observed that women with PCOS present higher serum concentration of TNF and C-reactive protein (CRP) as well as monocyte and lymphocyte circulating levels, and inflammatory infiltration in ovarian tissue [8]. This chronic inflammatory state is aggravated by obesity and hyperinsulinemia. There are studies describing the mutual impact of hyperinsulinemia and obesity, hyperandrogenism, and inflammatory state [50,51].

It is accepted that both obesity and hyperinsulinism are promoting molecular mechanisms implicated in higher androgens expression [52].

Data from many studies suggest not only correlative, but also causative association between higher activity of proinflammatory processes in adipose tissue and impair insulin metabolism, insulin resistance, and diabetes type 2 [53,54,55]. In obese women, there is an observed imbalance between classically activated macrophages (M1) and alternatively activated macrophages (M2). In these groups, there is higher concentration of M1 macrophages. Classically activated macrophages are characterized by expression of pro-inflammatory cytokines, CD11c surface expression, and nitric oxide synthase, while alternatively activated macrophages are characterized by anti-inflammatory cytokines expression, CD206, and arginase1 surface expression. That means that in obese individuals, there is a predominance of pro-inflammatory processes that causes systemic low-grade chronic inflammation [56,57,58,59,60,61,62].There are some studies that indicate a positive correlation between activity of classically activated macrophages and insulin resistance as well as increase in proinflammatory responses [61,63,64]. Activation of pro-inflammatory pathways such as JNK and TNF-kB signaling pathways causes higher production of pro-inflammatory cytokines, endothelial adhesion molecules, and chemotactic mediators that stimulate infiltration of monocytes in adipose tissue as well as differentiation into classically activated macrophages. These macrophages induce local, but also systemic pro-inflammatory status and impair insulin signaling [54,65,66,67,68]. What is important, some authors’ studies results show that normal-weight women with PCOS also have altered insulin signaling due to inflammatory processes in adipose tissue [69]. Meng Ch., in his meta-analysis, described the correlation between nitric oxide and PCOS by assessing nitrite levels [70].

Considering the fact that hyperinsulinemia and impair glucose tolerance are mayor factors aggravating the state of health of PCOS women and inflammatory process, it is important to notice that usage of particular insulin-sensitizers, such as inositol isoforms, gained increasing attention due to their safety profile and effectiveness [71,72]. Moreover, administration of inositol isoforms was effective in improving insulin sensitivity in both obese and lean PCOS women [71]. It is worth noting that it enhances the effect of metformin, but also clomiphene citrate, on fertility of PCOS women. Due to that fact, it can be especially useful in PCOS women who seeking pregnancy [72].

Obesity and insulin resistance are major factors of hyperglycemia states in PCOS women. Hyperglycemia can play a role in the inflammation process in PCOS women. Glucose is a main redox substrate of mononuclear circulating cells. In this process, formation of reactive oxygen species (ROS) is induced. That leads to activation of NF-kB, that is transcription factor, which is associated with expression of proinflammatory mediators, for example TNF or IL-6 [73]. Moreover, hyperglycemic state can promote secretion of steroidogenic molecules that can result in hyperandrogenemia [27]. Obesity is also associated with an elevated serum level of leptin, which has a proinflammatory effect. Hyperleptinemia promotes production of proinflammatory cytokines, such as TNF, IL-6, and IL-12 [74].

There is a lack of knowledge of how inflammation processes present in obese women are triggered. Recently, there have been published studies that analyze the potential molecular mechanisms that leads from obesity throughout increased inflammation to impair insulin metabolism and even diabetes type 2 [53,75,76]. Studies suggest that this process might involve dysregulation of fatty acids homeostasis, increased adipose tissue cells size and death, dysfunction of mitochondria, local hypoxia, as well as mechanical stress and endoplasmic reticulum stress [53,64,75,76,77,78]. Recent studies analyze the impact of mitochondrial dysfunction and it’s proinflammatory impact in PCOS women and its role in PCOS pathogenesis [47,48,79]. Mitochondrial dysfunction in this group of women is associated with increased ROS and oxidative stress that is observed in somatic cells and oocytes of women with PCOS. It is also associated with decreased mtDNA copy number and mutations in mtDNA regions encoding OXPHOS genes and tRNAs. All mechanisms mentioned may induce activation of apoptotic pathways in some cell types, including granulosa cells and acceleration of follicles and generation of healthy oocytes. All these molecular mechanisms are fully described by Cozzolino M. et al. [48].

Hyperandrogenemia, which is one of the key features of PCOS women, appears to inhibit glyoxylase-I activity. It plays an important role as enzymatic scavenging system for 2-oxoaldehydes, which also include major precursors of advanced glycation end products (AGE). This mechanism may lead to acceleration of deleterious effects of AGE deposition in PCOS women. Advanced glycation end products are metabolites of disrupted carbohydrate metabolism and are cytotoxic. Their accumulation in ovaries leads to oxidative stress and aberration in ovarian tissue structure. This process may lead to damage of all types of ovarian cells, but also to alteration of their function [79].

## 3. Endothelial Inflammation and Risk of Cardiovascular Disease

Various studies indicated relationship between PCOS and the higher risk of cardiovascular disease [3,80,81]. Hyperandrogenism and hyperinsulinemia, both common characteristics of PCOS patients, affect body fat distribution, hyperglycemia, dyslipidemia, and hypertension, and increase the risk of coronary artery disease. Vascular changes, such as endothelial dysfunction, increased arterial stiffness, and intima-media thickness, have been shown to be prevalent even among young women with PCOS [81]. Moreover, there are some studies confirming the association of PCOS, vascular disorders, and endothelial cell dysfunction independent of age, weight, and metabolic abnormalities.

Endothelial cell dysfunction leading to atherosclerosis may show as diminished capacity to nitric oxide (NO) production in response to injury by oxidized cholesterol, hyperglycemia, cigarette smoke, or hyperhomocysteinemia. Moreover, endothelial cell dysfunction may be triggered by inflammatory cytokines. Many players involved in oxidative stress, inflammation, and thrombosis were proposed as cardiovascular risk markers showing the endothelial cell damage in PCOS [82]. Those markers include asymmetric dimethylarginine (ADMA), C-reactive protein (CRP), homocysteine, plasminogen activator inhibitor-I (PAI-I),PAI-I activity, vascular endothelial growth factor (VEGF), advanced glycation and products (AGEs), lipoprotein a (Lp(a)), endothelin-1, ischemia-modified albumin (IMA) and urinary albumin excretion (UAE), interleukin-6 (IL-6),glypican-4, serum complement C3, copeptin, granzyme-B, pentraxin-3, hepcidin, interleukin -18 (IL-18), advanced oxidation protein products (AOPPs), and visfatin [83]. Even adolescents with PCOS have higher levels of CRP and PAI-1 than the control subjects [84].

Recently, pentraxin 3 (PTX3) arises as a new sensitive marker of endothelial dysfunction in PCOs patients [85,86]. PTX3, one of the acute phase proteins named pentraxins, is expressed locally in the site of inflammation. However, it still remains contradictory whether PTX3 correlates with BMI values, waist circumference, fat percentage, and insulin levels [87,88]. What is interesting is that PTX3 is not only synthesized by immune response or endothelial cells, but also by the granulosa cells in the cumulus oophorus, where it plays an important role in guaranteeing normal oocyte development and fertilization [89,90].

Imaging studies also confirm that endothelial damage occurs early in young women with PCOS. Angiography and increased coronary artery wall calcium determined by electron beam computed tomography suggests that PCOS subjects have more extensive coronary disease. Moreover, there is increased carotid intimal thickness on ultrasound in PCOS women [82].

Basic interventions focus at improving diet and promoting physical activity in order to modify the possible metabolic changes in PCOS. Insulin-sensitizers, mainly metformin, are proposed in subjects with hyperinsulinemia to ameliorate the metabolic profile. Additionally, N1-methylnicotinamide, a potential therapeutic agent, could modulate cardiometabolic markers involved in PCOS [83].

## 4. An Inflammatory Process in PCOS as a Risk Factor for Early Pregnancy

Most of the studies confirm that women with polycystic ovary syndrome are at increased risk of miscarriage after either spontaneous or assisted conception. Rates of early pregnancy loss were reported to be 30–50% [91,92]. The prevalence of PCOS among women with recurrent miscarriage appears to be high (30–82%) [93,94].

One of the main causes of early pregnancy loss in general is aneuploidy of the fetus. However, genetic analysis performed after early spontaneous miscarriages demonstrated that aneuploidy of the abortuses was significantly lower in PCOS (28.1%) than in the non-PCOS group (72.1%) [94]. Another study showed that early miscarriage rate in lean polycystic ovary syndrome women after euploid embryo transfer was higher compared to controls [95]. The researchers concluded that embryonic aneuploidy does not play a vital role in early spontaneous abortion in women with PCOS and that endometrial factors are more likely to be responsible for the increased risk of early pregnancy losses in PCOS patients.

Uterine molecular mechanisms of inflammatory modulation in normal pregnancy create an appropriate environment to receive the conceptus and orchestrate maternal immune toleration of the semi-allogeneic fetus. The control of the interaction between the placenta and membranes of the fetus is very complex. A uterine hyperinflammatory state in polycystic ovary syndrome may be responsible for significant pregnancy complications ranging from miscarriage to placental insufficiency [96].

It is possible that hyperinsulinemia and insulin resistance are responsible for reduced endometrial receptivity. Glycodelin is a major glycoprotein induced by progesterone and secreted from the secretory and decidual endometrium playing vital role in proper implantation. Glycodelin has been shown to induce apoptosis of monocytes. The decrease of glycodelin in the endometrium and in the serum is associated with increased rate of miscarriage. In PCOS women hyperinsulinemic state decreases the concentrations of circulating glycodelin [49]. Moreover, hyperinsulinemia may decrease expression of IGF binding protein-1, which is important in adhesion processes of feto-maternal interaction. Insulin-resistant states are also associated with increased plasma plasminogen activator inhibitor-1 concentration and a hypofibrinolytic state being additional independent risk factor for miscarriage in PCOS [92,93]. There are some studies suggesting that metformin treatment during pregnancy in PCOS patients may reduce the risk of early pregnancy loss [97].

The research in the field of inflammatory balance of the uterus during early pregnancy shows new possible correlations and targets for future action. It has been suggested that abnormal expression of the inflammasome NLRP3 and a decoy receptor D6 are involved in the pathogenesis of higher rates of miscarriage in PCOS [96]. Inflammasomes are high molecular weight, intracellular multiprotein complexes enabling caspase-mediated processing of some cytokines and representing the first line of defense against microbial invasion and cellular stress. The inflammatory response is mediated by several chemokines, which in turn promote leukocyte recruitment to sites of inflammation. D6 decoy receptor binds most of the cytokines and is responsible for chemokine gradients in tissues.

Full comprehension of the pathogenesis of the miscarriages of euploid fetuses in PCOS patients has not been reached. The understanding of uterine molecular modulation of inflammation could provide new therapeutic targets for pregnancy disorders.

## 5. Summary

According to known data, inflammatory mediators are higher in PCOS patients. The continuous release of inflammatory markers is associated with long-term metabolic complications and higher risk of coronary heart disease. Although exact mechanisms are not fully understood yet, there are numerous studies that underline mutual impact of obesity and insulin resistance in increased inflammation, suggesting probabilistic initiating or modulating influence of these states on PCOS pathogenesis. There are also studies indicating a relationship between higher concentration of androgens and white blood cells count, so that chronic low grade inflammation may be mediated not only through adiposity, but also through androgens concentration. Further studies are needed to establish those relationships, especially the association of new polymorphisms of genes which play significant roles in the pathophysiologic mechanism implicated in the inflammatory process in PCOS.

## Figures and Tables

**Table 1 ijms-22-03789-t001:** Inflammatory markers in polycystic ovary syndrome (PCOS) patients.

Authors Ref.	Year of Publication	Study Design	Main Findings
Kelly, C.C., Lyall, H., Petrie, J.R. et al. [14]	2001	Clinical study	elevated CRP concentration in PCOS patients versus control group
Orio, F., Palomba, S., Casella, T. et al. [17]	2005	Clinical study	elevated CRP and WBC concentration in PCOS patients versus control group
Souza Dos Santos A.C., Soares, N.P. et al. [16]	2015	Clinical study	elevated CRP concentration in PCOS patients versus control group
Tola, E.N., Yalcin, S.E., Dugan, N. [15]	2017	Clinical study	elevated CRP concentration in PCOS patients versus control group WBC-distributed homogenously between PCOS and control group
Rudnicka, E., Kunicki, M., Suchta, K. et al. [10]	2020	Clinical study	elevated CRP and WBC concentration in PCOS patients versus control group
Escobar Morreale H.F., Laque-Ramirez, M., Gonzalez, F. [18]	2011	Systematic review and meta-analysis	elevated CRP concentration in PCOS patients versus control group, no statistically significant difference in IL-6 and TNF-αin PCOS patients versus control group
Toulis, K.A., Goulis, D.G., Mintziori, G. et al. [19]	2011	Systematic review and meta-analysis	elevated CRP concentration in PCOS patients versus control group no statistically significant difference in IL-6 and TNF-αin PCOS patients versus control group
Kaya, C., Pabuccu, R., Berker, B. et al. [22]	2010	Clinical study	elevated IL-18 concentration in PCOS patients versus control group
Escobar-Morreale, H.F., Botella-Carretero, J., Villuendas, G. et al. [23]	2004	Clinical study	elevated IL-18 concentration in PCOS patients versus control group
Glintborg, D., Andresen, M., Richelsen, B. et al. [24]	2009	Clinical study	elevated MCP-1 concentration in PCOS patients versus control group
Franik, G., Sadlocha, M., Madej, P. et al. [12]	2020	Clinical study	decreased concentration of omentin -1 levels in PCOS patients versus control group
Hu, W., Qiao, J., Yang, Y. et al. [25]	2011	Clinical study	elevated CRP and MCP-1 concentration in PCOS patients versus control group
Papolou, O., Livadas, S., Karachalios, A. et al. [36]	2015	Clinical study	elevated WBC concentration in PCOS patients versus control group
Herlihy, A.C., Kelly, R.E., Hogan, J.L. et al. [37]	2011	Clinical study	elevated WBC concentration in PCOS patients versus control group
Phelan, N., O’Connor, A., Tun, K. et al. [38]	2013	Clinical study	elevated WBC concentration in PCOS patients versus control group
Garg, D., Merhi, Z. [41]	2016	Clinical study	elevated AGEs concentration in PCOS patients versus control group
Merhi, Z. [42]	2014	Review	elevated AGEs concentration in PCOS patients versus control group
Diamanti-Kandarakis, E., Piperi, C., Kalofoutis, A. et al. [43]	2005	Clinical study	elevated AGEs and RAGE concentration in PCOS patients versus control group
Zhang, J., Bao, Y., Zhou, X. et al. [47]	2019	Review	mitochondrial dysfunction plays a role in pathogenesis of PCOS by increasing ROS and oxidative stress
Cozzolino, M., Seli, E. [48]	2020	Review	mitochondrial dysfunction plays a role in pathogenesis of PCOS by increasing ROS and oxidative stress
Uysal, S., Isik, A.Z., Eris, S. et al. [49]	2015	Clinical study	decreased endometrial glycodelin (induce apoptosis of monocytes) expression in PCOS women with miscarriage

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
