# Peer review of "Chronic Low Grade Inflammation in Pathogenesis of PCOS"

_ijms, 2021, doi:10.3390/ijms22073789_

Round 1

Reviewer 1 Report

I read with great interest the manuscript, which falls within the aim of this Journal. In my honest opinion, the topic is interesting enough to attract the readers’ attention. Nevertheless, authors should clarify some points and improve the discussion, as suggested below.

Authors should consider the following recommendations:

  • Manuscript should be further revised in order to correct some typos and improve style.
  • Accumulating evidence suggests that one of the most important mechanisms of PCOS pathogenesis is the insulin-resistance. For this reason, the use of insulin-sensitizers, such as inositol isoforms, gained increasing attention due to their safety profile and effectiveness. Authors may better discuss this point, taking to account these recent articles: PMID: 28835764; PMID: 32396844.

Reviewer 2 Report

The manuscript describes selected elements of PCOS pathogenesis and the development of its cardiovascular complications, focusing on inflammatory markers.

Main comments:

  1. The manuscript does not contain the authors' own thoughts on the interpretation of the discussed research. Also, the summary does not include the authors' own thoughts.
  2. The manuscript does not cite one of the first studies on the effect of nutritional status and PCOS on circulating chronic inflammation markers

Eur J Obstet Gynecol Reprod Biol. 2007 Aug;133(2):197-202. d

  1. The lack is discussion of studies describing new inflammatory marker pentraxin-3 in PCOS women and its role as a surrogate marker of endothelial dysfunction. Numerous studies were published in recent years:

Front Immunol. 2018 Nov 29;9:2808. 

Gynecol Obstet Invest. 2014;78(3):173-8.

Int J Clin Exp Med. 2014 Oct 15;7(10):3512-9. 

Eur J Endocrinol. 2014 Feb 4;170(3):401-9.

Int J Endocrinol. 2020 Sep 2;2020:1380176.

Zhejiang Da Xue Xue Bao Yi Xue Ban. 2020 Oct 25;49(5):637-643.

Scand J Clin Lab Invest. 2019 Oct;79(6):419-423.

  1. Page 1. Type II diabetes should be changed to type 2 diabetes. In addition PCOS is a results of obesity and insulin resistance not a cause.
  2. Page 2. It is not true that the studies clearly show the concentration of pro-inflammatory cytokines in women with PCOS with comparable BMI values.
  3. Page 3. It is not true that TNF-alpha is probably not related with the pathogenesis of the syndrome per se, but might have a modulating effect on its phenotype. The effect of pro-inflammatory cytokines, including TNF-alpha, on LH and FSH secretion has been described.
  4. Page 6 There are no citations after the sentence “There are studies describing mutual impact of hyperinsulinemia and obesity, hyperandrogenism and inflammatory state”.
  5. Page 6 The sentence “What is important, some authors’ studies results show that also normal-weight women with PCOS have increased adipose tissue resistance[69].” Is unclear. What is increased adipose tissue resistance?
  6. Page 7 As was mentioned above among endothelial dysfunction markers the PTX3 is not mentioned.
  7. Page 8 The sentence “An inflammatory process in PCOS- risk factor for early pregnancy?” should be rewritten because in this form it is not a question.
  8. Page 8. The sentence “lean polycystic ovary syndrome women” should be rewritten to normal weight women with PCOS.
  9. References should be unified according to journal guidelines.

Round 2

Reviewer 2 Report

After revision I don't have any comments